# Magnetic Nanoparticles as In Vivo Tracers for Alzheimer's Disease

**Bhargy Sharma [1,*]** and **Konstantin Pervushin [2,*]**

1    School of Materials Science & Engineering, Nanyang Technological University, Singapore 639798, Singapore
2    School of Biological Sciences, Nanyang Technological University, Singapore 637551, Singapore
*    Correspondence: bhargy001@e.ntu.edu.sg (B.S.); kpervushin@ntu.edu.sg (K.P.); Tel.: +65-6790-6421 (B.S.);
     +65-6514-1916 (K.P.)

**Abstract:** Drug formulations and suitable methods for their detection play a very crucial role in the development of therapeutics towards degenerative neurological diseases. For diseases such as Alzheimer's disease, magnetic resonance imaging (MRI) is a non-invasive clinical technique suitable for early diagnosis. In this review, we will discuss the different experimental conditions which can push MRI as the technique of choice and the gold standard for early diagnosis of Alzheimer's disease. Here, we describe and compare various techniques for administration of nanoparticles targeted to the brain and suitable formulations of nanoparticles for use as magnetically active therapeutic probes in drug delivery targeting the brain. We explore different physiological pathways involved in the transport of such nanoparticles for successful entry in the brain. In our lab, we have used different formulations of iron oxide nanoparticles (IONPs) and protein nanocages as contrast agents in anatomical MRI of an Alzheimer's disease (AD) brain. We compare these coatings and their benefits to provide the best contrast in addition to biocompatibility properties to be used as sustainable drug-release systems. In the later sections, the contrast enhancement techniques in MRI studies are discussed. Examples of contrast-enhanced imaging using advanced pulse sequences are discussed with the main focus on important studies in the field of neurological diseases. In addition, T1 contrast agents such as gadolinium chelates are compared with the T2 contrast agents mainly made of superparamagnetic inorganic metal nanoparticles.

**Keywords:** Alzheimer's disease; iron-oxide nanoparticles; magnetic resonance imaging; early detection; magnetic contrast

---

## 1. Alzheimer's Disease

Alzheimer's disease (AD) is the most common form of dementia, comprising 70–80% of the total cases. With the overall global burden of dementia estimated to reach USD 2 trillion and more than 80 million cases by the year 2030 with currently a new diagnosis of AD every 65 s, there is a dire need for a cure or treatment which has evaded researchers until now [1]. The gold standard for AD diagnosis is the identification of insoluble plaques or tangles detected in the postmortem brain through histological staining. The key pathological events in the disease progression, such as Amyloid-β (Aβ) aggregation and phosphorylation of tau protein, can begin decades before the appearance of the first clinical symptoms. Such findings have rightfully shifted the focus towards early diagnosis of these hallmarks or biomarkers [2]. The diagnostic modalities for brain imaging include positron electron tomography (PET), computed tomography (CT), and magnetic resonance imaging (MRI). Currently, these techniques are used in conjunction with neuropsychological testing after the appearance of first clinical symptoms to estimate the progression of the disease. MRI is considered safe compared to the other techniques since there is no need for ionizing radiation or any radioactive tracers to be involved at

any point during the scanning process. MRI provides excellent soft-tissue contrast which can be used for diagnosis of disease pathology in the brains of probable AD patients. It is a suitable tool for longitudinal scans as a standard clinical imaging modality for follow-up after therapeutic interventions. However, the key to developing therapeutics for AD is the early detection of possible pathological changes, long before the onset of cognitive decline. For such elusive anatomical differences, the inherent tissue contrast is not enough. Several molecules are developed over the decades to provide enhanced contrast in MRI, which can be used as tracers for early diagnosis of amyloidogenic changes in the brain. One critical consideration for the development of such molecules is their ability to enter the target brain regions.

## 2. Diagnostic Tracers for Targeting Brain Tissue in Alzheimer's Disease

Antibodies and aptamers are developed for specific receptors on brain endothelial cells to be used as AD-associated biomarkers [3]. Small molecules such as Congo red, Curcumin, and Methoxy-XO4 can probe Aβ plaques with high binding affinity through hydrophilic functional groups and be used as fluorescent tracers. However, they need to be carried into the brain regions due to their poor bioavailability. Liposomes are used as micro- and nanoscale drug carriers to take therapeutics across the blood-brain barrier (BBB) [4]. They can incorporate hydrophobic, hydrophilic or lipophilic compounds due to their amphiphilic nature. They assimilate in the body through adsorptive-mediated endocytosis by interacting with cellular membranes. Higher concentrations are required in the dosage to cause a significant impact. Furthermore, non-specific uptake and potential cytotoxicity restrict their use as nanocarriers to the brain. The biochemical properties of these liposomes are modified through surface engineering to make them suitable transporters.

Similarly, there are preparations of polymers, micelles, dendrimers and quantum dots which can be modified into brain-targeting molecular tracers. Most of these nanoparticle tools require combinations of different functional properties in addition to select functional groups which allow for their use as detection tools. Ultra-small inorganic nanoparticles such as iron-oxide nanoparticles (IONPs) can be functionalized with desired properties and biologics in addition to fluorescent or optical tags which gives them an edge over their other counterparts (Figure 1). While many of them are developed to bind radioactive labels to be used as PET tracers, several formulations of nanoparticles with a magnetic core are being used as contrast enhancers for MRI to overcome the half-life and safety concerns associated with other more invasive imaging modalities.

## 3. Modes of Brain Targeting for Detection of Alzheimer's Disease

Methods of administration to reach the brain regions are a crucial point to heed during the process of tracer development. In addition to most of the administered probes getting sequestered out of the circulation before reaching brain targets, they may attract non-specific groups to bind due to their versatile outer layers leading to the formation of corona, leading to unexpected effects elsewhere in the body. With most of these systems developed as AD probes which are administered through a systemic route, crossing the blood-brain barrier (BBB) is a crucial hurdle to overcome. The BBB comprises endothelial cells interconnected by tight junctions which control the transport of molecules into the brain. Some studies suggest that roughly a quarter of pores in the brain extracellular space junctions have more 100 nm pore size [5].

The traditional methods of probe administration include intravascular, intraperitoneal or dermal injections. However, brain targeting can be achieved more specifically by non-invasive modes of intranasal or oral administration. Overall, the different ways of probe administration to target the brain can be mainly categorized into two groups as described below.

### 3.1. Across the Blood-Brain Barrier

The BBB, which is a physical semi-permeable interface between the blood vessels and the brain parenchyma, restricts the diffusion of most molecules and prevents them from entering the brain. The overall size of less than 500–600 Da and the highly lipophilic surface of the nanoparticles allow

them to cross the BBB through facilitated diffusion [6]. Small polar drug particles lesser than 600 Da utilize surface receptors or carriers for their transport across the barrier [6]. It has been argued that the lipophilic molecules with a molecular weight below 400 Da are the ideal candidates for targeting brain regions due to the pore size at the BBB junctions and the physical properties of the barrier [7]. However, this limitation is being pushed by new research focused on modifying the surface properties of nanoparticles to overcome this requirement by conferring better properties for interaction with the formidable BBB. Some nanoparticle formations administered intravenously, such as angiopep-conjugated poly (ethylene glycol)-copoly (ε-caprolactone) nanoparticles, have been shown to cross the BBB in healthy mice [8]. It is generally assumed that the integrity of the BBB breaks down during AD progression, which is seen as a possibility for easier access to the brain for the injected therapeutics [9]. However, imaging studies do not support this hypothesis as the BBB is seen to be intact in AD mouse models showing other disease symptoms. Over the last years, disruption of BBB is induced prior to administration of brain targeting drugs through systemic injections. The breakdown of the BBB at any point significantly increases the risk of toxicity to the microvasculature in the brain, which is often detected as imaging abnormalities like edema using MRI. Functionalization with specific biomolecules can promote transport across the BBB. Novel fusion technologies allow the delivery of larger particles such as proteins or drugs across the BBB by binding small peptide-mimicking monoclonal antibodies or "trojan horse"molecules for specific receptors on the BBB to facilitate the transport across the barrier into the brain [10].

Materials like gold and silica work well for transport, even without functionalization [5]. Transcytosis movement of hydrophilic molecules via endothelial cells is adsorptive-mediated for positively charged nanoparticles and receptor-mediated for proteins. Retrograde transport, monocyte mimicking and BBB breakdown strategies have also been explored with success for only up to 20 nm particles to seek assistance in the entry of targeted agents into the brain. In addition to a meager fraction of injected agent reaching the brain, intravenous administration of nanoparticles also suffers from the formation of "corona" due to non-specific interactions with molecules in the blood possibly leading to surface modifications. Silica-coated nanoparticles can cross the BBB even without any further functionalization and have been successfully used as intranasal agents for brain targeting. For particles up to 200 nm in size, the endocytosis is mediated by clathrin-coated pits while particles larger than 600 nm are taken up through caveolae [5]. However, even after delivery across the BBB, immunotoxicity effects have been a significant concern. Wadghiri et al. reported the use of Polyethylene glycol (PEG) for coating iron oxide nanoparticles, which not only showed a decrease in the effects of immunotoxicity but also improved the BBB permeability [11].

### 3.2. Bypassing the Blood-Brain Barrier

Diagnostic and therapeutic probes can enter the choroid plexus in the brain through the blood-Cerebrospinal fluid (CSF) barrier, which is anatomically different from the BBB and not as impermeable. The biochemical properties of these molecules determine whether they can move to targeted brain regions or get washed off by the CSF, which turns over entirely in 4–5 h. At the same time, the CSF-to-blood-plasma ratio of certain biologics is used as a biomarker for both BBB intactness as well as possible AD progression [12]. Intrathecal injection directly into the ventricles is an invasive method to target biologics which can ensure most of the injected sample enters the brain. These probes can then diffuse out into the brain tissue through the ventricles. However, diffusion or bulk flow to specific brain tissue depends on the propensity of these biologics towards the specific region. Many of the biomolecules are restricted to ventricular CSF flow before being sequestered out of the body through the blood circulation due to lack of receptors or the ability to diffuse [13]. Leptomeningeal space occupied by the CSF can enhance the diffusion of nanoparticles through different brain regions.

Intranasal administration is a way to bypass the BBB to transport the nanoparticles olfactory pathway or trigeminal nerve pathway instead of the systemic circulation. Previously used intranasal injections tested for drugs face a limitation of active concentration reaching the brain due to

comparatively lesser surface adsorption through olfactory lobes [5]. Deposition of Aβ in olfactory bulbs which precedes their presence in isocortex and hippocampus provides strong support to methods like intranasal administration [14]. Studies have demonstrated that the addition of peptides or proteins allows more accumulation of probes into the cerebrum and cerebellum, previously shown using fluorescent probes. The smaller core size (<200 nm) and hydrophilic coatings are formulated to help the nanoparticles avoid sequestration by the reticuloendothelial system and achieve a longer circulation time [5]. Intranasal administration of FDA approved acetylcholinesterase inhibitors for AD, i.e., rivastigmine, galantamine, and donepezil has also shown higher bioavailability for brain targeting.

## 4. Magnetic Resonance Imaging in Alzheimer's Disease

MRI is the preferred technique for non-invasively monitoring the progression of a disease or impact of treatment on account of its superior soft-tissue contrast. It can be used to track the biomarkers or exogenously administered tracers for AD diagnosis. MRI brain scans utilize the magnetization transfer between the macromolecules and protons in the bulk water. Typical brain MRI includes whole-brain multi-planar images over the brain volume, magnetic resonance angiography, diffusion imaging and chemical shift imaging. These techniques traditionally utilize the inherent contrast of the brain, which arises from differences in T1 and T2 relaxation times of different types of tissue with very high values for the CSF. The T1 or longitudinal relaxation time ranges from few milliseconds to several seconds in biological tissues. The T2 or transverse relaxation can take place independently due to static local field disturbances or accompany the T1 relaxation. It is usually shorter than T1. The magnetic inhomogeneity or susceptibility artefacts in tissue lead to an observed relaxation time lesser than the actual value of T2, known as T2*. Hemorrhages and calcifications in the tissue mostly affect this difference between the two relaxation times. When the AD clinical symptoms are reported, the amyloid plaques and vascular amyloids can be directly detected through MRI. However, such studies using the inherent contrast require ultra-high-field scanners and require very long scanning times which are not compatible with clinical imaging. Contrast-enhanced MRI, which can reduce the T1 or T2 relaxation times in the presence of exogenous tracers, has tremendous potential for early detection of amyloidosis. Paramagnetic nanoparticles comprising lanthanide elements such as gadolinium ($Gd^{3+}$) or transition metals such as manganese ($Mn^{2+}$) affect the T1 and T2 relaxation times, and have been traditionally used as positive contrast agents in MRI, preferentially fastening the T1 recovery with contrast-enhanced regions appearing bright or "hyperintense" on T1-weighted images. On the other hand, ferromagnetic and superparamagnetic iron oxide contrast agents preferentially increase the relaxation leading to darker appearance or "hypointensity" for contrast-enhanced tissues. IONPs typically reduce T2 and T2* relaxation times and are mostly imaged through T2*-weighted techniques [15].

A spin-echo (SE) MRI pulse sequence uses a 90° radiofrequency (RF) pulse followed by 180° refocusing pulse which reverses the dephasing to generate a train of echoes. SE sequences usually generate MRI images with high signal-to-noise ratio, which is attributed to their insensitivity to susceptibility variations and magnetic field inhomogeneity. The echo time (TE) and repetition time (TR) impact the image contrast with T1-weighting resulting from sequences using short TR and TE. Another type of commonly used MR pulse sequences for imaging of MRI contrast agents are the gradient recalled echo (GRE) sequences. A gradient-echo signal is acquired by a single RF pulse to get the MRI image through GRE. Once the RF pulse is turned off, the switching of the gradient field causes the precessing spins to refocus, leading to the generation of an echo. Such sequences are mostly used to evaluate the extent of iron deposits in the brain due to their sensitivity to tissue susceptibility artefacts, magnetic field inhomogeneities and chemical shift variations [16]. GRE sequences provide the benefit of quick scanning times with shorter TE combined with smaller TR. Fast Low Angle Shot (FLASH) first proposed by Haase et al. is a versatile GRE technique with RF excitation at low flip angles which reduces the saturation effects and produces disruption of the transverse coherence, also known as spoiling, which also allows for T1-weighted imaging as well [17]. Over the years, this technique has undergone significant transformation leading to more sophisticated pulse sequences which are

preferred for imaging the IONPs [18]. Susceptibility-weighted imaging (SWI) enhances the sensitivity of GRE sequences to T2* effects by allowing phase mapping. However, such scans take longer and lead to lower quality images due to motion disturbances. Nevertheless, GRE pulse sequences are the preferred choice for imaging the contrast changes due to IONPs as hypointensity in the brain MRI images. The loss in signal is higher with longer TE, causing an increase in T2* sensitivity due to greater dephasing. Thus, T2* weighting is achieved through lower flip angles and longer TE and TR.

## 5. Iron Oxide Nanoparticles as Magnetic Resonance Imaging Probes

Gadolinium-based contrast agents are usually available as biocompatible chelates of $Gd^{3+}$ to prevent any competitive inhibition of biological processes involving calcium ions. Although several Gd-based contrast agents are commercially available, their clinical use has been suspended, citing the risk of nephrogenic systemic fibrosis due to its accumulation in tissues. IONPs are seen as alternative MRI contrast agents much safer than Gd. The IONPs are superior to gadolinium in the sense that they are required in lesser concentrations to bring about the same relaxation changes in the surrounding water molecules. Due to their smaller size, they adopt superparamagnetism with each molecule acting as a single magnetic domain, and hence get magnetized to a greater extent compared to the same number of Gd-DTPA nanoparticles which are commonly used as T1 contrast agents [19]. IONPs can be functionalized with desired motifs to render specific characteristics such as biocompatibility and solubility. When combined with their small size, it allows them to target brain regions which might otherwise not be accessible. While the nephrotoxicity of gadolinium nanoparticles is a cause for concern, it is suggested that the IONPs get integrated into the biological iron reserves after a few days in the system [20]. Hence, they can be used for longitudinal studies of follow-up detection tools after therapeutic interventions. FDA approved IONPs have core sizes of 50–200 nm, good biodistribution and biocompatibility, and many were approved for clinical use, including Resovist®, Clariscan®, and Feridex® before taken out from markets [21]. They are conventionally synthesized by co-precipitating a mix of $Fe^{2+}$ and $Fe^{3+}$ salts such as $FeCl_2.4H_2O$ and $FeCl_3.6H_2O$ in alkaline media to produce pure $Fe_3O_4$ nanoparticles with constant stirring in an inert environment [22]. The nanoparticles precipitate and are magnetically harvested, followed by vacuum drying at high temperatures. In addition to magnetic properties, the selection of ideal IONPs depends on other tunable properties such as bioconjugation, reduced size, stability, solubility, and biocompatibility under physiological conditions, to name a few. They can be conferred specific properties by selectively binding to functional motifs of antibodies, sugars, proteins or small biomolecules. $Fe^{3+}$ ions boast higher *r1* relaxivity which shows the potential of the IONPs to be developed as both T1 and T2 contrast agents. In most pre-clinical studies, the presence of IONPs in the tissue is confirmed through histological staining such as Perl Blue after MRI.

Iron oxide-based magnetic contrast agents directly affect the spin-spin relaxation, also known as T2 relaxation, which is the decay of transverse magnetization in the neighboring protons. Dextran-coated IONPs, for example, Ferumoxides, have been used for MR relaxometry studies in mammalian cells to compare the presence or absence of iron using the Carr-Purcell-Meibom-Gill pulse sequences [23]. These coated IONPs are generally biodegradable and can be dissolved due to low pH within lysosomes or endosomes within a few days, or phagocytosis by macrophages. Various formulations of iron nanoparticles have been developed as magnetic resonance contrast agents by coating the IONPs with different molecules. IONPs injected intravenously were shown to reach the brain through choroid plexus seen in mice induced for neuroinflammation [24]. PEG on the surface also helps the nanoparticles penetrate the BBB, which can overcome the requirement of a co-injection with hyperosmolar solutions such as mannitol to open the crossing temporarily [25]. Biocompatible surface coatings such as silica and dopamine facilitate the retention of the IONPs for a longer time to allow for their detection in longitudinal scans before the need for re-injection to attain enough MRI contrast. In addition to standard iron oxide core composition, several studies have shown better magnetic properties using a metal complex composition in the core. Better relaxivity and higher saturation

magnetization were achieved by using ferromagnetic Zinc-Manganese core to grow $Fe_3O_4$ shell on them, rendering superparamagnetic properties [21]. Chitosan-coated $MnFe_2O_4$-$Fe_3O_4$ also showed better relaxivity and low cytotoxicity compared to only $Fe_3O_4$ or $MnFe_2O_4$ compositions [26].

Ferritin is an intracellular iron storage protein composed of 24 subunits comprising heavy and light chains which together form a 480 kDa spherical shell. The naturally occurring protein holoferritin consists of ferrihydrite ($5Fe_2O_3·9H_2O$) in comparison to the magnetite ($Fe_3O_4$) or maghemite ($\gamma$-$Fe_2O_3$) cores incorporated artificially into engineered ferritin nanocages for higher MR contrast. IONPs enclosed with protein shells of recombinant human heavy-chain ferritin protein have been previously used as detection agents for tumor tissues, though their potential as MR active agents was overlooked [27]. The benefits of using protein-cage assemblies as the outer layer are the homogeneity in size and lesser propensity for aggregation compared to inorganic nanoparticles [28]. Ferritin nanocages have been utilized as contrast agents by direct encapsulation of inorganic iron within the protein shell. The ferrtin mutant K150A and R151A in *Archaeoglobus fulgidus* (AfFtn-AA) is engineered to self-assemble into closed-pore nanocages assembly from dimers [29]. Engineered ferritin nanocages have higher *r1* as well as *r2* relaxivities as compared to the gadolinium complexes. In addition to their use as MRI contrast agents, the tunable assembly and disassembly of these nanocages can be utilized to encapsulate protein or drug cargo for the development of AD therapeutics [30]. The superparamagnetic nature of the magnetic nanoparticles allows them to be administered in lower concentrations compared to other contrast agents and easily detectable in MRI. There is also innovation in substituting naturally occurring iron-binding proteins such as ferritin as non-toxic magnetic nanoparticles.

## 6. Magnetic Resonance Imaging Probes for Alzheimer's Disease Therapeutics

Anti-A$\beta$ antibodies are conjugated to IONPs to allow for the visualization of amyloid structures in the diseased brain [15]. NU-4 antibodies that specifically target toxic soluble A$\beta$ oligomers successfully conjugated to PEG-coated IONPs were observed to bind (24–48)-unit A$\beta$ oligomers in AD mice brains following intranasal administration [31]. These antibodies were also detected using MRI when probed on probable AD human brain slices and 5xFAD mouse model. IONPs dually functionalized with the amyloid-binding dye Congo Red and the antioxidant Rutin could accumulate with amyloid plaques in AD transgenic mice brain and alleviate the memory deficits (Figure 2A) [32]. PEG-PLA coated IONP conjugated with Curcumin could label A$\beta$ plaques in Tg2576 AD mice brain slices (Figure 2B) [25]. IONPs were conjugated to A$\beta$(1-42) peptides due to their inherent affinity towards the amyloid plaques to detect aggregation in the diseased brain. These probes were detected in the AD-associated brain regions, such as cortex and hippocampus, of the transgenic mice after intravenous injections in MRI scans and immunolabeling studies (Figure 3) [11]. Other hallmarks of AD can also be used for diagnosis using magnetic nanoparticles. For example, IONPs functionalized with anti-tau antibodies can label tau proteins in the blood plasma of AD patients under the influence of alternating magnetic fields [33].

While the multi-functionality of IONPs allows for their use as theragnostic agents, the choice of the therapeutic molecules bound to these contrast agents is also very crucial. The antibodies and synthetic molecules have an inherent risk of triggering immunogenic reactions in the body. Many non-proteolytic proteins, such as Lipocalin-type Prostaglandin D Synthase (L-PGDS), clusterin, $\alpha$B-crystallin, and $\alpha$2-Macroglobulin, act as extracellular chaperones and can inhibit the aggregation of A$\beta$ [34,35]. These endogenous human proteins act as chaperones for misfolded A$\beta$ peptides and show disaggregase activity towards the insoluble A$\beta$ aggregates by breaking them down into monomers. Therefore, extracellular chaperones can provide better alternatives to antibodies for formulations of the AD tracers. L-PGDS, which is the second most abundant protein in the CSF, binds to the A$\beta$ monomers and fibrils with high binding affinity ($K_D$ ~ nM range) [36]. L-PGDS prevents the primary and secondary nucleation of A$\beta$(1-40) peptides, thereby preventing their fibrillation [37]. It was previously reported that L-PGDS could sequester more than twenty aggregated proteins from the insoluble extract from the brains of AD patients [37]. We recently showed that the nanoprobes with L-PGDS covalently conjugated to AfFtn-AA nanocages can be used to detect A$\beta$ in the brain of the

triple transgenic AD mouse model. These probes co-localize with the brain amyloids within 24 h as observed through high-resolution MRI (submitted, to be published). These conjugates could target the AD-associated brain regions, such as hippocampus and cortical nuclei, upon non-invasive intranasal administration. Conjugate probes of L-PGDS with other biocompatible IONPs with PEG, silica and dopamine coatings were also capable of inhibiting Aβ aggregation in vitro and were not toxic to neuronal cells. To the best of our knowledge, the L-PGDS-AfFtn-AA conjugates are the first all-protein conjugates with a magnetic core that can successfully bind and breakdown mature amyloid fibrils. They show tremendous potential as diagnostic and therapeutic tools for MRI in AD brains. L-PGDS can bind to the low-density lipoprotein receptor in the brain, which facilitates the transport of conjugated nanoparticles for brain targeting across the BBB [38]. The ability of L-PGDS to bind and disassemble Aβ at sub-stoichiometric concentrations renders an added advantage to its development as a non-invasive intranasally administered tracer. Similar to L-PGDS-IONP conjugates, other Aβ chaperones can be chemically linked to magnetic nanoparticles to successfully produce effective AD tracers.

## 7. The Future Outlook for Alzheimer's Disease Tracers

Current limitations for the use of IONPs as MRI diagnostics and therapeutics stem from a lack of dedicated MRI pulse sequences with optimal signal-to-noise ratios and scan times. Additionally, the lack of optimized protocols of MRI image registration and analysis for mouse brain images also poses a hindrance for extensive pre-clinical animal studies to explore the full potential of drug-conjugated IONPs as suitable diagnostic and therapeutic tools. Lack of consistencies in the different AD mouse models and the age differences between the appearance of AD hallmarks can be attributed to the absence of a standard protocol for conducting such studies. Commercial production of clinically relevant IONP-based contrast agents such as Ferumoxytol®, Resovist®, Feridex®, and Sinarem® ceased due to preferential T2* shortening and lack of demonstrable utility. Tunable nanoparticles which can simultaneously modulate T1 and T2 contrast can help overcome the ambiguities that arise in MRI images. Synthesis of such probes is gaining traction in recent years, and a breakthrough towards AD prevention can be made by combining their diagnostic properties with endogenous therapeutic molecules.

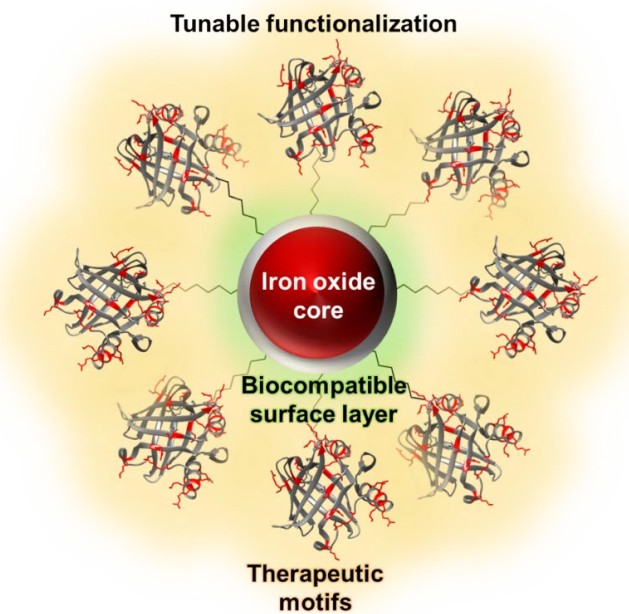

**Figure 1.** Schematic for tunable magnetic tracers for Alzheimer's disease theragnostic tools. The magnetic core of the nanoparticles provides the detection capability for magnetic resonance imaging (MRI) with the functionalized biologics bestowing the therapeutic capability.

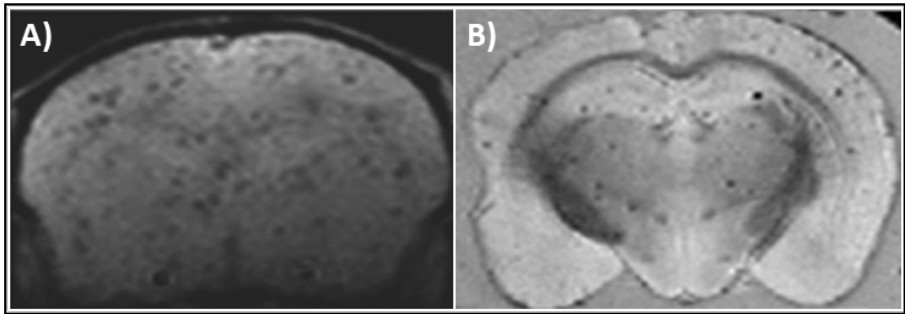

**Figure 2.** Coronal MRI images showing detection of functionalized iron oxide nanoparticles (IONPs) in Alzheimer's disease (AD) mouse brain regions. (**A**) In vivo T2* weighted image in APPswe/PS1dE9 transgenic AD mouse brain 12 h after intravenous injection of Congo red/Rutin conjugated IONPs at 7T MRI. Study by Hu et al. [32]. (**B**) Ex vivo T2* image of 18-month old Tg2576 transgenic AD mouse brain 4 h after intravenous injection of curcumin conjugates polyethylene glycol (PEG)-supported IONPs at 7T MRI. Study by Cheng et al. [25]. Images adapted with permission.

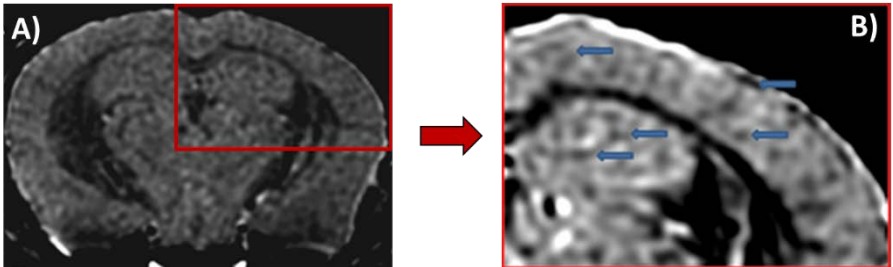

**Figure 3.** T2*-weighted image of 14-month old APP/PS1 transgenic mice after intravenous femoral injection of IONPs conjugated with PEG and Aβ1-42. (**A**) Coronal slice of whole brain at 7T MRI with 100 μm isotropic resolution, with highlighted region, including the cortex and the hippocampus, shown with higher magnification in (**B**). Arrowheads mark some of the hypointense regions indicative of IONP-PEG-Aβ1-42 localization corroborated with immunolabeling studies. Images from Wadhgiri et al. [11].

**Author Contributions:** Conceptualization, B.S. and K.P.; writing—original draft preparation, B.S. and K.P.; writing—review and editing, B.S. and K.P.; supervision, K.P.; funding acquisition, K.P. Both authors have read and agreed to the published version of the manuscript.

**Funding:** This work was funded by Grant from Ministry of Education of Singapore to Konstantin Pervushin, grant number M4012175.

**Conflicts of Interest:** The authors declare no conflict of interest.

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
