# Peer review of "Magnetic Nanoparticles as In Vivo Tracers for Alzheimer’s Disease"

_magnetochemistry, doi:10.3390/magnetochemistry6010013_

Round 1

Reviewer 1 Report

The authors review the main strategies related to contrast agents in order to detect Alzheimer's disease through imaging techniques. The content of this paper is well argued and organized, despite being a little short in relation to the vast amount of literature in this field.
It is a bit surprising that a review of contrast agents does not present material referring to magnetic resonance images that allow the reader to visualize what is described in the text. This reviewer would recommend the use of additional imaging material

Reviewer 2 Report

The manuscript presents an interesting work. 

My main comment is related to last paragraph of section 6, mainly because I do not understand the English. For example "and dissagregase towards..", "Once such abundant brain protein Lipocalin-285 type Prostaglandin D Synthase (L-PGDS) binds to Aβ monomers and fibrils with high affinity"? All the paragraph has been properly re-written.
